# The feasibility of SARS-CoV-2 surveillance using wastewater and environmental sampling in Indonesia

Indah K. Murni[1,2]*, Vicka Oktaria[1,3]*, Amanda Handley[4,5], David T. McCarthy[6], Celeste M. Donato[4,7], Titik Nuryastuti[8], Endah Supriyati[9], Dwi Astuti Dharma Putri[1], Hendri Marinda Sari[1], Ida Safitri Laksono[1,2], Jarir At Thobari[1], Julie E. Bines[4,7,10]

1 Faculty of Medicine, Center for Child Health–Pediatric Research Office, Public Health and Nursing, Universitas Gadjah Mada, Yogyakarta, Indonesia, 2 Faculty of Medicine, Child Health Department, Public Health and Nursing, Universitas Gadjah Mada, Yogyakarta, Indonesia, 3 Faculty of Medicine, Department of Biostatistics, Epidemiology and Population Health, Public Health and Nursing, Universitas Gadjah Mada, Yogyakarta, Indonesia, 4 Enteric Diseases Group, Murdoch Children's Research Institute, Parkville, Victoria, Australia, 5 Medicines Development for Global Health, Southbank, Victoria, Australia, 6 Department of Civil Engineering, Environmental and Public Health Microbiology Lab (EPHM Lab), Monash University, Clayton, Victoria, Australia, 7 Department of Paediatrics, The University of Melbourne, Parkville, Australia, 8 Faculty of Medicine, Department of Microbiology, Public Health and Nursing, Universitas Gadjah Mada, Yogyakarta, Indonesia, 9 Faculty of Medicine, Center for Tropical Medicine, Public Health, and Nursing, Universitas Gadjah Mada, Yogyakarta, Indonesia, 10 Department of Gastroenterology and Clinical Nutrition, Royal Children's Hospital Melbourne, Parkville, Victoria, Australia

* indah.kartika.m@ugm.ac.id (IKM); vicka.oktaria@ugm.ac.id (VO)

## Abstract

### Background

Wastewater-based epidemiology (WBE) surveillance as an early warning system (EWS) for monitoring community transmission of SARS-CoV-2 in low- and middle-income country (LMIC) settings, where diagnostic testing capacity is limited, needs further exploration. We explored the feasibility to conduct a WBE surveillance in Indonesia, one of the global epicenters of the COVID-19 pandemic in the middle of 2021, with the fourth largest population in the world where sewer and non-sewered sewage systems are implemented. The feasibility and resource capacity to collect samples on a weekly or fortnightly basis with grab and/or passive sampling methods, as well as to conduct qualitative and quantitative identification of SARS-CoV-2 ribonucleic acid (RNA) using real-time RT-PCR (RT-qPCR) testing of environmental samples were explored.

### Materials and methods

We initiated a routine surveillance of wastewater and environmental sampling at three predetermined districts in Special Region of Yogyakarta Province. Water samples were collected from central and community wastewater treatment plants (WWTPs), including manholes flowing to the central WWTP, and additional soil samples were collected for the near source tracking (NST) locations (i.e., public spaces where people congregate).

**Data Availability Statement:** The dataset can be found in this link: https://doi.org/10.6084/m9.figshare.19824445.

**Funding:** This Project was funded by the Global Innovation Fund and PATH (PATH.org). The Global Investment Fund had no involvement in study design, data collection or analysis and PATH participated in study design and reviewing the draft manuscript, but had no role in data collection or analysis, writing of the manuscript or the decision to submit it for publication.

**Competing interests:** The authors have declared that no competing interests exist.

## Results

We began collecting samples in the Delta wave of the COVID-19 pandemic in Indonesia in July 2021. From a 10-week period, 54% (296/544) of wastewater and environmental samples were positive for SARS-CoV-2 RNA. The sample positivity rate decreased in proportion with the reported incidence of COVID-19 clinical cases in the community. The highest positivity rate of 77% in week 1, was obtained for samples collected in July 2021 and decreased to 25% in week 10 by the end of September 2021.

## Conclusion

A WBE surveillance system for SARS-CoV-2 in Indonesia is feasible to monitor the community burden of infections. Future studies testing the potential of WBE and EWS for signaling early outbreaks of SARS-CoV-2 transmissions in this setting are required.

## Introduction

Understanding the full extent of the Coronavirus Disease (COVID-19) pandemic is a major public health challenge. Traditional epidemiological indicators which are based on the number of confirmed clinical cases and deaths due to COVID-19 disease have potential biases and limitations. The capacity for timely diagnosis using laboratory tests may be limited, particularly in low- and middle- income countries (LMICs) during epidemic wave. Incidence rates based on hospitalization data lag behind the incidence of infection in the community and lack of representativeness for identification of cases who do not access care, have non-serious illness, or are asymptomatic.

People infected with SARS-CoV-2 shed the virus in stool independently of gastrointestinal symptoms and therefore viral ribonucleic acid (RNA) can be detected in environmental wastewater, containing excreta from infected people and sewerage treatment plants [1–4]. Public health surveillance using wastewater is now well established and has been used to monitor communities for the presence of poliovirus, antimicrobial resistant enteric bacteria, and drugs of abuse, e.g. opioids [5–7]. It has been postulated that routine monitoring for the presence of SARS-CoV-2 in wastewater may be useful in detecting an existing or predicting a new potential epidemic [6, 8].

Studies reporting the detection of SARS-CoV-2 RNA in wastewater have been predominantly limited to high-income countries such as Australia, the United States, Japan and a number of European countries. To date, only a few studies have detected the genetic material of SARS-CoV-2 in wastewater from LMICs, including studies from Argentina, Brazil, Ecuador, India, Pakistan, and South Africa [9–20]. The lack of formal sewerage systems in LMICs, particularly in impoverished areas and informal settlements, has posed a major challenge for SARS-CoV-2 surveillance using wastewater. It is also in these communities where epidemiological surveillance using rates based on disease case capture and death are problematic. The adaptation of environmental surveillance methods suitable for use in LMICs provides an opportunity to monitor community transmission and inform the public response to SARS-CoV-2 and other future pandemic infections.

This short communication describes the assessment of the feasibility of conducting SARS-CoV-2 surveillance using wastewater and environmental sampling in Indonesia. The aim was

to provide a proof of concept for the use of wastewater and environmental surveillance to monitor the community burden of SARS-CoV-2 infection in Indonesia.

## Materials and methods

### General information on wastewater systems and challenges in Indonesia

In Indonesia, a high proportion of the population is not connected to a sewerage system. In the capital city of Jakarta, a city with a population of over 10 million, it is estimated that only 2% of households are connected to a reticulated sewerage system, with >95% of wastewater leaking into agricultural fields, rivers, and other groundwater sources [21].

We established the first Indonesian wastewater-based SARS-CoV-2 epidemiology surveillance program in Special Region of Yogyakarta province, one of the regions with the highest number of COVID-19 cases during the Delta wave. In the Special Region of Yogyakarta province, only 25,294 households (6% population serviced) are connected to a formal reticulated sewerage system. There are two types of wastewater treatment plants (WWTPs) systems in operation in the province: (a) the central WWTP (*Instalasi Pengolahan Air Limbah Sewon/ IPAL* Sewon, Bantul) managed by the provincial government and (b) community WWTPs (IPAL community) that are independently managed by each local community, in addition to individual septic tanks. The service coverage of IPAL Sewon in the Special Region of Yogyakarta province includes 13 of the 14 sub-districts in the Yogyakarta city, 4 of the 17 sub-districts in the Sleman district and 3 of the 17 sub-districts in the Bantul district. Community WWTPs are used in some suburban areas due to the lack of capacity of the central WWTPs to service their needs and the terrain of the region that does not allow passive gravitational flow.

### SARS-CoV-2 surveillance on wastewater and environmental sampling in Indonesia (SWESP study)

Routine wastewater-based epidemiology (WBE) surveillance (i.e., testing of sewerage and wastewater sites, and waterways) and testing of soil was initiated in three of five districts in the Special Region of Yogyakarta province (Yogyakarta city, Sleman and Bantul districts, **Fig 1**). Two districts were not included due to practical challenges, such as the geography and relatively sparse population. Identification and mapping of the infrastructure of the wastewater system (formal and informal) at provincial and district level was conducted prior to commencing the study. We selected six sub-districts from Yogyakarta city as these areas have the highest coverage of the formal central wastewater system and samples may be considered more representative to the broader community, two from Sleman district, and the remaining two from Bantul district. Within the total of ten sub-districts, we also selected 12 clustered communities that were served by small community WWTPs. Each community WWTP served between 50–150 households.

We collected samples using either the grab or passive sampling methods. Wastewater from manholes was collected by immersing a ~500 mL bottle into the water to a depth of around 20–30 cm until the bottle was filled, allowing about 1 cm of air. Recreational water was collected using a 2 L bottle using a similar grab method. Bottles were pre-labelled with sample specific barcodes. A torpedo-style passive sampler with multiple entry points (front, top, sides, and bottom) [22, 23] was used to collect samples from septic tanks, rivers, and the central and community WWTPs. Passive samplers were retrieved 24 hours after deployment. Soil samples (20 g) were collected using zip lock bags. Within four hours of collection, samples were transferred on ice at 2–8˚C [24] to the Microbiology laboratory at the Universitas Gadjah Mada Special Region of Yogyakarta, Indonesia.

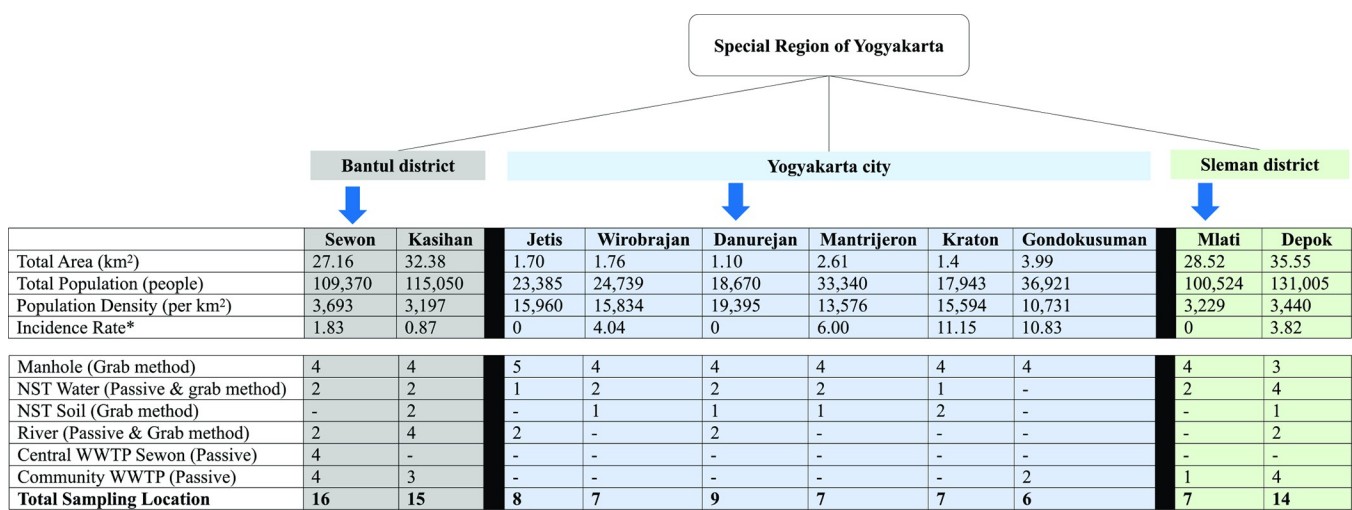

**Fig 1. Flowchart of sample strategy.** We selected ten sub-districts from three out of five districts in Special region of Yogyakarta Province (Yogyakarta city, Bantul, and Sleman districts). Samples from three sub-districts were taken weekly (identified by blue arrows), while others were taken fortnightly. Detailed type and number of samples in each sub-district are illustrated in the figure.

## Laboratory methods for wastewater and environmental samples

The wastewater samples, passive samplers and soil samples were stored in the 4°C fridge upon arrival until the sample processing. Samples of wastewater (50 mL) or recreational water (1000 mL) were filtered through a 47 mm diameter, 0.45 μm pore size, cellulose nitrate high flow electronegative membrane (Sartorius, Germany). This filtration process was performed immediately (<2 hours) once the samples were received at the laboratory. The collection bag containing the soil samples was thoroughly mixed. In a 2 mL tube, 0.25 grams of soil and 2 mL of DNA/RNA Shield solutions (Zymo Research, USA) were added. The passive samplers were opened, and the filter membrane and q-tips were collected.

All of the processed samples from wastewater samples, passive samplers and soil samples were stored at -80°C until the RNA extraction and reverse-transcription quantitative real-time PCR (RT-qPCR) analysis.

The RNA was extracted from samples using the QIAGEN RNeasy PowerMicrobiome Kit (QIAGEN, Germany) following manufacturer's instructions with the exception of replacing the supplied beads with PowerBead Tubes-Garnet beads (QIAGEN, Germany). For every batch of samples processed, a negative extraction controls and internal control (MS2 bacteriophage) as supplied in the PerkinElmer SARS-CoV-2 Nucleic Acid Detection Kit (RUO) (PerkinElmer) were included in the RNA extraction process to monitor the RNA extraction performance.

To detect the SARS-Cov2 RNA, a RT-qPCR was conducted using the SARS-CoV-2 Real-time RT-PCR Assay (PerkinElmer, US) and synthetic SARS-CoV-2 RNA Control 1-MT007544.1 (Twist Bioscience, Australia) as the standard curve. The kit is a multiplex assay using primers and probes targeting the Nucleocapsid (N) gene and open reading frame 1ab (ORF1ab) region of SARS-CoV-2. RT-qPCR assays were performed using two replicates of 5 μL RNA template, with a total reaction volume of 30 μL and a total 45 cycles of amplification. The quantification of the samples was calculated using the synthetic SARS-CoV-2 RNA Control 1-MT007544.1 (Twist Bioscience, Australia) as a standard curve, according to the manufacturer's instruction. The RT-qPCR assay was performed as described by the manufacturer's instruction using the LightCycler 96 instrument (Roche, Germany).

In order to report the actual value of SARS-CoV-2 RNA, we calculated the recovery effi-ciency. In each qPCR run, multiple SARS-CoV-2 RNA controls, a MS2 phage control (to determine the RNA recovery efficiency and as internal control) of different known concentra-tions and a negative control were included.

The limit of detection (LOD) for the RT-qPCR assay was determined by the analysis of 10 replicates for each dilution of the synthetic SARS-CoV-2 RNA Control 1-MT007544.1 (Twist Bioscience, Australia) analyzed and was defined as the lowest number of copies of the N gene target and ORF1ab gene that could be detected in 80% of the replicates tested. The LOD was expressed as the lowest detectable concentration of the N gene target and ORF1ab gene in sam-ple based on the equivalent volume of sample analyzed in each RT-qPCR assay, not adjusting for any potential loss through the processing of the sample or any potential inhibition of the RT-qPCR assay [25]. All assays were performed at Microbiology laboratory at the Universitas Gadjah Mada, Special Region of Yogyakarta, Indonesia.

### Ethics

The SWESP study obtained ethics approval from the Medical and Health Research Ethics Committee (MHREC), Faculty of Medicine, Public Health and Nursing, Universitas Gadjah Mada DR. Sardjito General Hospital, Indonesia (KE/FK/0426/EC/2021, KE/FK/0514/EC/2022). Written or verbal consent was not applicable for this study as we did not collect data from individual participants.

## Results

### Feasibility of WBE surveillance

The average time from sample collection to availability of the RT-qPCR results was a mean of 64 hours, including the filtration time (3 to 4 hours), RNA extraction (2 to 3 hours), and RT-qPCR quantification analysis (3 hours). Both weekly and fortnightly sample collections were practical to conduct. A key challenge was the delay in the importation of critical reagents and consumables exacerbated by the COVID-19 pandemic. As the UGM laboratory is also the cen-tral clinical laboratory, priority for the analysis of clinical samples resulted in a delay in waste-water analysis during major clinical peaks in incidence. Initial trials of deployment of the passive samplers were required to limit damage or loss due to difficulties with positioning and securing samplers. We defined criteria for reliable deployment that considered locations with solid ground to safely access, ideally in an inconspicuous position, and using a strong pole or tree to secure the sampler. To avoid samplers being removed we labeled samplers with signs of warning and explanation such as "Sample for Research by Universitas Gadjah Mada and Yog-yakarta Government".

### The detection and positivity rates of SARS-CoV-2 RNA

Sample collection commenced on the 27th of July 2021 during the Delta wave of the COVID-19 pandemic in Indonesia. During the 10-week sampling period, a total of 544 samples were collected with 54% (296/544) of all samples testing positive for SARS-CoV-2 RNA. The median of cycle threshold (Ct) values for positive N and ORF1ab gene results was 35.1 (IQR: 32.1–36.9) and 33.9 (IQR: 30.1–35.9), respectively. The highest positivity rate was for manhole sam-ples (74%, 191/258 samples, **Fig 2**) and the lowest was for soil samples (3%, 2/60 samples, **Fig 2**). The temporal changes in rates of sample positivity correlate with the number of confirmed cases in the community as illustrated in **Fig 3**. The highest positivity rate of 77%, was obtained for samples collected in July 2021 during week 1 of sample collection and decreased to 25% by

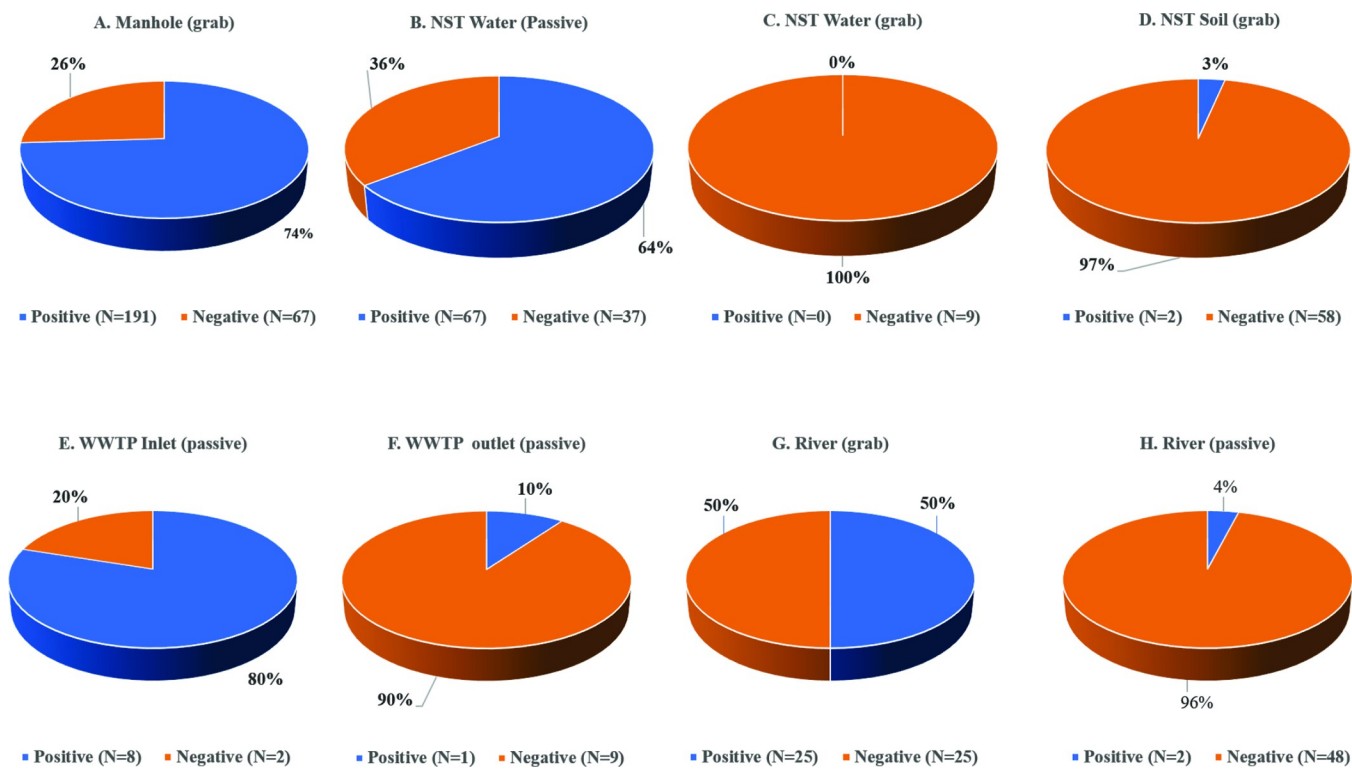

**Fig 2. Nucleocapsid (N) gene positivity by sample types.**

the end of September 2021 (corresponding to week 10 of sample collection), reflecting a decreased detection rate correlating with a decrease in the incidence of reported COVID-19 clinical cases in the community.

The N gene was identified in 74% (191/258) of sewage samples (grab method), 64% (67/104) of near source tracking (NST) water samples (passive sampling method), 50% (25/50) of river samples (grab method), and 3% (2/60) of NST soil samples. This finding was consistent with the ORF1ab gene target but with a higher proportion of soils samples being positive (8%, 5/60) for the ORFlab gene as compared to the N gene (3%).

## Discussion

We successfully demonstrated that WBE surveillance for SARS-CoV-2 RNA was feasible in Indonesia and reflected the SARS-CoV-2 clinical burden in the community. The high level of positivity of SARS-CoV-2 RNA in the environment in Indonesia suggests a considerable public health burden and may represent asymptomatic or mild cases that did not access health facilities for testing. Manholes consistently showed higher positivity rates in comparison with river and soil samples. Although river and soil samples showed lower positivity rates, the data are useful to complement the WBE surveillance data particularly in regions where connection to a formal sewerage system is limited. This combination of sampling strategies provides additional insights into the prevalence and distribution of COVID-19 within the community.

In Special Region of Yogyakarta province, many households are not connected to the IPAL Sewon. This may be because they were built after the IPAL Sewon infrastructure was established and therefore have no connection to the IPAL pipes. Other households were not connected due to technical reasons, such as in lower altitudes and terrain that does not support

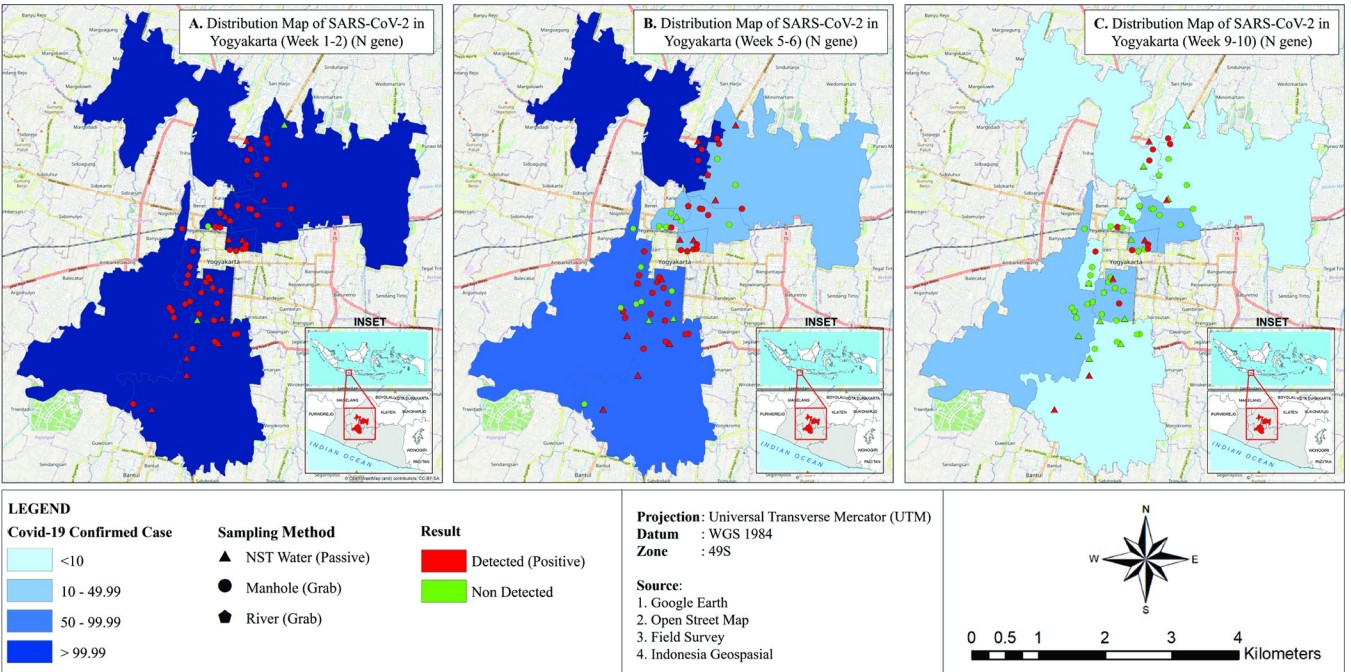

**Fig 3. Distribution maps of SARS-CoV-2 in Special Region of Yogyakarta province, comparing detection targeting N gene to community confirmed cases.** (A) In week 1–2 of the sample collection. (B) In week 5–6 of the sample collection. (C) In week 9–10 of the sample collection. Community COVID-19 confirmed cases were represented by blue color, the lighter the fewer cases. Detected cases in sampling locations were represented by red colored dots/triangles/ pentagons, while non-detected cases were represented by green colored dots/triangles/pentagons. With circles denoting manholes, pentagons denoting river and triangles denoting NST water.

passive gravitational flow of wastewater to the central WWTP. However, in this study we managed to collect samples from community WWTPs and septic tanks from NST sites to capture communities that were not served by the central WWTP.

Although we found that both weekly or fortnightly collection frequency with grab and/or passive sampling collection methods as feasible, weekly collections were preferred in order to provide real-time data to inform the public health response. The laboratory capacity to conduct qualitative (positive/negative) and quantitative identification of SARS-CoV-2 RNA in the environmental samples (wastewater and soil) were also feasible although some pre-processing procedures need to be conducted prior to the RT-qPCR procedure (i.e., wastewater filtration and soil homogenization). There were challenges in providing real-time results during peak COVID-19 outbreaks due to overburdened staff and limited access to equipment, and therefore, ideally WBE surveillance should be integrated into the routine surveillance programs with dedicated staff. Additionally, the availability of imported reagents has delayed laboratory analysis during periods of high output. Local epidemiological data describing the distribution of COVID-19 cases (symptomatic and asymptomatic) with laboratory confirmed positive tests for SARS-CoV-2 infections by sub district, on a weekly basis, were available to compare with the findings from WBE surveillance. However, data analysis to link environmental and community data remains challenging and needs further exploration.

Despite efforts, there remain practical limitations of WBE surveillance in LIMCs. It is likely that wastewater sampling of the reticulated sewerage system reflects the more modern and affluent sector of the city and may not provide meaningful insights into the presence of SARS-CoV-2 infection within the broader community. Most of the city and rural areas manage human effluent via septic tanks, pit latrines or by open defecation with subsequent

contamination of surface water and rivers. Therefore, to understand the distribution of SARS-CoV-2 RNA in environments that reflect the presence of community infections with fragmented wastewater infrastructures, NST sites, and in places where people publicly congregate were selected. These sites include permanent dwellings (apartment and flats), temporary living places (hotels), public spaces (traditional markets, town squares, mosques, and a public swimming pool), rivers, working spaces (both office and factory), and COVID-19 shelters (facilities which are designated as temporary quarantine shelters for people testing positive for COVID-19). This WBE approaches using NST may allow detection of targeted clusters for whom rapid action may reduce or prevent the risk of larger outbreaks within the community [26].

It has been proposed that WBE surveillance has the potential to act as an early warning system (EWS) for COVID-19 outbreaks [27–32]. This should be conducted in collaboration with the public health authorities to enable the timely follow up of positive detections by strategies such as contact tracing, strengthening health protocols, or implementing a community lockdown. This could be broadly implemented across the community or in a targeted response depending on the local context and level of concern. For instance, if SARS-CoV-2 RNA is detected (positive result) in the sewerage sample in an area where there had consistently been no detections (negative result), then a lockdown or mass screening could be implemented in the area drained by the sewerage system; or if the result is taken from a closed community (e.g., Boarding school), contact tracing within the community should be conducted immediately.

## Conclusions

In conclusion, an environmental surveillance system for SARS-CoV-2 in Indonesia is feasible and can be used to monitor the community burden of SARS-CoV-2 infection. However, future research is needed to explore its potential to act as an EWS for the early identification of SARS-CoV-2 outbreaks within a community, especially in regions with limited access to clinical testing. Although the sewer infrastructure of wastewater systems is quite limited in Indonesia, an expanded sampling approach based on the local context and including NST can support an effective SARS-COV-2 surveillance program.

## Acknowledgments

We would like to thank the wastewater treatment plant team, field assistants and laboratory team for doing sampling collection and laboratory works. We thank Rizka Dinari for providing editorial assistance.

## Author Contributions

**Conceptualization:** Indah K. Murni, Vicka Oktaria, Amanda Handley, David T. McCarthy, Celeste M. Donato, Titik Nuryastuti, Julie E. Bines.

**Data curation:** Amanda Handley, Endah Supriyati, Dwi Astuti Dharma Putri, Hendri Marinda Sari.

**Formal analysis:** Indah K. Murni, Vicka Oktaria, Dwi Astuti Dharma Putri.

**Funding acquisition:** Amanda Handley, Julie E. Bines.

**Investigation:** Indah K. Murni, Vicka Oktaria, Amanda Handley, David T. McCarthy, Celeste M. Donato, Julie E. Bines.

**Methodology:** Indah K. Murni, Vicka Oktaria, Amanda Handley, David T. McCarthy, Celeste M. Donato, Titik Nuryastuti, Endah Supriyati, Julie E. Bines.

**Project administration:** Amanda Handley, Endah Supriyati, Dwi Astuti Dharma Putri, Hendri Marinda Sari.

**Resources:** Indah K. Murni, Vicka Oktaria, David T. McCarthy, Titik Nuryastuti, Julie E. Bines.

**Supervision:** Indah K. Murni, Vicka Oktaria, Titik Nuryastuti, Ida Safitri Laksono, Jarir At Thobari, Julie E. Bines.

**Validation:** Indah K. Murni, Vicka Oktaria, Amanda Handley, David T. McCarthy, Titik Nuryastuti, Endah Supriyati, Julie E. Bines.

**Writing – original draft:** Indah K. Murni.

**Writing – review & editing:** Indah K. Murni, Vicka Oktaria, Amanda Handley, David T. McCarthy, Celeste M. Donato, Titik Nuryastuti, Endah Supriyati, Dwi Astuti Dharma Putri, Hendri Marinda Sari, Ida Safitri Laksono, Jarir At Thobari, Julie E. Bines.

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
