## [Decision Letter · Decision Letter 0]

1 Jul 2022

PONE-D-22-14874The feasibility of SARS-CoV-2 surveillance using wastewater and environmental sampling in IndonesiaPLOS ONE

Dear Dr. Murni,

Thank you for submitting your manuscript to PLOS ONE. After careful consideration, we feel that it has merit but does not fully meet PLOS ONE’s publication criteria as it currently stands. Therefore, we invite you to submit a revised version of the manuscript that addresses the points raised during the review process. As pointed out by one of the reviewers, please describe the data of qPCR.

We look forward to receiving your revised manuscript.

Kind regards,

Etsuro Ito

Academic Editor

PLOS ONE

Journal Requirements:

"We would like to acknowledge PATH for supporting the study and reviewing the draft

manuscript. Learn more at PATH.org"

"This Project was funded by the Global Innovation Fund and PATH. The Global Investment Fund had no involvement in study design, data collection or analysis and PATH participated in study design, but had no role in data collection or analysis, writing of the manuscript or the decision to submit it for publication."

Reviewers' comments:

Reviewer's Responses to Questions

**Comments to the Author**

1. Is the manuscript technically sound, and do the data support the conclusions?

Reviewer #1: Yes

Reviewer #2: Partly

2. Has the statistical analysis been performed appropriately and rigorously? 

Reviewer #1: Yes

Reviewer #2: No

3. Have the authors made all data underlying the findings in their manuscript fully available?

Reviewer #1: Yes

Reviewer #2: No

4. Is the manuscript presented in an intelligible fashion and written in standard English?

Reviewer #1: Yes

Reviewer #2: Yes

5. Review Comments to the Author

Reviewer #1: This paper presents a feasibility study of using wastewater-based epidemiology for early detection of SARS-CoV-2 in low- and middle in come countries. The study is based on 10-week period data collected from key locations. The results show the highest rate of 77% and the lowest rate of 25% in positivity. Overall, I have a positive impression over this paper. I see that it makes a contribution in three ways: First, it focuses on low- and middle-income countries, which may often be disadvantaged in terms of receiving early notification of the virus spread. Second, it provides a practical proof of concept showing evidence of how this approach may be used to track drastic changes in the positivity over a period of time. Third, it provides meaningful discussion on how this approach may can be used in control policy.

Reviewer #2: This is an important and relevant paper that looks at the feasibility of wastewater surveillance in a low- and middle-income country. Specifically, in a city with a population of over 10 million where it is estimated that only 2% of households are connected to a reticulated sewerage system, with >95% of wastewater leaking into agricultural fields, rivers, and other groundwater sources. However, there are main concerns that need to be addressed.

Main concerns

1- The paper describes how samples were collected and analyzed but there is no data showing RT-qPCR results. The authors should follow: “The MIQE Guidelines: Minimum Information for Publication of Quantitative Real-Time PCR Experiments” (https://academic.oup.com/clinchem/article/55/4/611/5631762) when reporting their results.

2- The results provided by the authors appear correct:

- higher positivity rate in wastewater corresponds with clinical data

- higher positivity rate for samples from wastewater treatment plans vs rivers and soil

However, with out the data described above these results can’t be validated.

Minor concerns

1- There are a few typos in the manuscript that suggest that perhaps the authors did not upload the latest version.

2- The resolution of the figures should be improved.

3- A map showing the geographical location of the samples would be helpful to interpret results.

6. PLOS authors have the option to publish the peer review history of their article (what does this mean?). If published, this will include your full peer review and any attached files.

Reviewer #1: No

Reviewer #2: No

---

## [Author Response · Author response to Decision Letter 0]

14 Aug 2022

Response to Reviewers [PONE-D-22-14874]

The feasibility of SARS-CoV-2 surveillance using wastewater and environmental sampling in Indonesia

Editor's comment:

As pointed out by one of the reviewers, please describe the data of qPCR.

Response to the Editor:

We thank you for your interest and positive comments on this paper. In this revised version, we have described the data or RT-qPCR following the MIQE Guidelines: Minimum Information for Publication of Quantitative Real-Time PCR Experiments, as pointed out by the Reviewer #2.

Response to reviewers:

Reviewer #1:

This paper presents a feasibility study of using wastewater-based epidemiology for early detection of SARS-CoV-2 in low- and middle-income countries. The study is based on 10-week period data collected from key locations. The results show the highest rate of 77% and the lowest rate of 25% in positivity. Overall, I have a positive impression over this paper. I see that it makes a contribution in three ways: First, it focuses on low- and middle-income countries, which may often be disadvantaged in terms of receiving early notification of the virus spread. Second, it provides a practical proof of concept showing evidence of how this approach may be used to track drastic changes in the positivity over a period of time. Third, it provides meaningful discussion on how this approach may can be used in control policy.

Response to reviewer #1:

We greatly appreciate your positive comments on this paper.

Reviewer #2:

This is an important and relevant paper that looks at the feasibility of wastewater surveillance in a low- and middle-income country. Specifically, in a city with a population of over 10 million where it is estimated that only 2% of households are connected to a reticulated sewerage system, with >95% of wastewater leaking into agricultural fields, rivers, and other groundwater sources. However, there are main concerns that need to be addressed.

Main concerns

1- The paper describes how samples were collected and analyzed but there is no data showing RT-qPCR results. The authors should follow: “The MIQE Guidelines: Minimum Information for Publication of Quantitative Real-Time PCR Experiments” (https://academic.oup.com/clinchem/article/55/4/611/5631762) when reporting their results.

Response to reviewer 2:

Thank you for pointing out The MIQE Guidelines for reporting RT-qPCR experiments. We provided the information on RT-qPCR experiments such as details of the samples, nucleic acid extraction, reverse transcription, and qPCR target information under the section ‘Laboratory methods for wastewater and environmental samples’ (page 8-10, line 147-186). We also have amended the paper to describe the data and the RT-qPCR as described in the MIQE Guidelines. Furthermore, we have added the median of RT-qPCR cycle threshold data during the 10-week of sample collection for both the N and ORF1ab genes.

“The median of cycle threshold (Ct) values for positive N and ORF1ab gene results was 35.1 (IQR: 32.1 – 36.9) and 33.9 (IQR: 30.1 – 35.9), respectively.” (results section, page 11, line 212-214)

2- The results provided by the authors appear correct:

- higher positivity rate in wastewater corresponds with clinical data

- higher positivity rate for samples from wastewater treatment plans vs rivers and soil

However, without the data described above these results can’t be validated.

Response to reviewer #2:

We found that the changed trends of the positivity rate detected using N and ORF1ab genes were in alignment with confirmed cases in the community. In order to support this result, we attached the distribution maps of SARS-CoV-2 in Yogyakarta, comparing the detection of SARS-CoV-2 targeting N gene to confirmed community cases (Fig 3). 

The distribution maps illustrate three time points, i.e., week 1-2 (Fig 3A), week 5-6 (Fig 3B) and week 9-10 (Fig 3C). The areas were colored with different shades of blue to depict the number of confirmed COVID-19 cases in community, in which the darker color means the higher confirmed COVID-19 cases. The sampling locations were marked using circles (manholes), pentagons (river) and triangles (NST water) which are colored red if detected as positive or colored green if non-detected. In week 1-2, the COVID-19 cases in the community were high (dark blue), almost all sampling locations were positive for COVID-19 (only two green dots/triangles were shown). As the confirmed cases decreased through time, in week 9-10 the areas were shaded with light blue and the number of sampling locations detected as positive was lower. In addition, we revised the relevant results section to summarize this finding, as below.

“The temporal changes in rates of sample positivity correlate with the number of confirmed cases in the community as illustrated in Fig 3. The highest positivity rate of 77%, was obtained for samples collected in July 2021 during week 1 of sample collection and decreased to 25% by the end of September 2021 (corresponding to week 10 of sample collection), reflecting a decreased detection rate correlating with a decrease in the incidence of reported COVID-19 clinical cases in the community.” (results section, page 11, line 216-221)

We also added the summary comparing positivity rate between sample types (in results section, page 11, line 214-216) and pie charts of positivity rates of each sample type (Fig 2, see below) to support the finding that river and soil samples showed the lowest positivity rate.

“The highest positivity rate was for manhole samples (74%, 191/258 samples, Fig 2) and the lowest was for soil samples (3%, 2/60 samples, Fig 2).” (results section, page 11, line 214-216)

Minor concerns

1- There are a few typos in the manuscript that suggest that perhaps the authors did not upload the latest version.

Response to reviewer #2:

Thank you for pointing out this matter, we have corrected the typos and ensured that the submitted version is the latest version of our work.

2- The resolution of the figures should be improved.

Response to reviewer #2:

We have re-read the guideline for figures and reviewed our figures following the PLOS ONE requirements using the suggested PACE tool.

3- A map showing the geographical location of the samples would be helpful to interpret results.

Response to reviewer#2:

The newly attached maps of Fig 3 showing the distribution of SARS-CoV-2 in Yogyakarta have addressed this feedback. This figure is a geographical map of Yogyakarta province. As explained previously, we colored the sub-districts included in this study based on the number of confirmed COVID-19 cases. We also marked the sampling location and gave different symbols to each sample type, i.e., circle (manholes), pentagon (river) and triangle (NST water).

---

## [Decision Letter · Decision Letter 1]

6 Sep 2022

The feasibility of SARS-CoV-2 surveillance using wastewater and environmental sampling in Indonesia

PONE-D-22-14874R1

Dear Dr. Murni,

We’re pleased to inform you that your manuscript has been judged scientifically suitable for publication and will be formally accepted for publication once it meets all outstanding technical requirements.

Kind regards,

Etsuro Ito

Academic Editor

PLOS ONE

Reviewers' comments:

Reviewer's Responses to Questions

**Comments to the Author**

1. If the authors have adequately addressed your comments raised in a previous round of review and you feel that this manuscript is now acceptable for publication, you may indicate that here to bypass the “Comments to the Author” section, enter your conflict of interest statement in the “Confidential to Editor” section, and submit your "Accept" recommendation.

Reviewer #2: All comments have been addressed

2. Is the manuscript technically sound, and do the data support the conclusions?

Reviewer #2: Yes

3. Has the statistical analysis been performed appropriately and rigorously? 

Reviewer #2: Yes

4. Have the authors made all data underlying the findings in their manuscript fully available?

Reviewer #2: Yes

5. Is the manuscript presented in an intelligible fashion and written in standard English?

Reviewer #2: Yes

6. Review Comments to the Author

Reviewer #2: The authors have addressed all the comments. I believe the figures would benefit from higher resolution.

7. PLOS authors have the option to publish the peer review history of their article (what does this mean?). If published, this will include your full peer review and any attached files.

Reviewer #2: No

---

## [Editor Report · Acceptance letter]

5 Oct 2022

PONE-D-22-14874R1 

The feasibility of SARS-CoV-2 surveillance using wastewater and environmental sampling in Indonesia 

Dear Dr. Murni:

I'm pleased to inform you that your manuscript has been deemed suitable for publication in PLOS ONE. Congratulations! Your manuscript is now with our production department. 

Kind regards, 

on behalf of

Prof. Etsuro Ito 

Academic Editor

PLOS ONE